# Non-linear System Identification from Partial Observations via Iterative Smoothing and Learning

## Abstract

System identification is the process of building a mathematical model of an unknown system from measurements of its inputs and outputs. It is a key step for model-based control, estimator design, and output prediction. This work presents an algorithm for non-linear offline system identification from partial observations, i.e. situations in which the system's full-state is not directly observable. The algorithm presented, called SISL, iteratively infers the system's full state through non-linear optimization and then updates the model parameters. We test our algorithm on a simulated system of coupled Lorenz attractors, showing our algorithm's ability to identify high-dimensional systems that prove intractable for particle-based approaches. We also use SISL to identify the dynamics of an aerobatic helicopter. By augmenting the state with unobserved fluid states, we learn a model that predicts the acceleration of the helicopter better than state-of-the-art approaches.

## 1 Introduction

The performance of controllers and state-estimators for non-linear systems depends heavily on the quality of the model of system dynamics (Hou & Wang, 2013). System-identification tackles the problem of learning or calibrating dynamics models from data (Ljung, 1999), which is often a time-history of observations of the system and control inputs. In this work, we address the problem of learning dynamics models of partially observed, high-dimensional non-linear systems. That is, we consider situations in which the system's state cannot be inferred from a single observation, but instead requires inference over a time-series of observations.

The problem of identifying systems from partial observations arises in many robotics domains (Punjani & Abbeel, 2015; Cory & Tedrake, 2008; Ordonez et al., 2017). Though we often have direct measurements of a robot's pose and velocity, in many cases we cannot directly observe relevant quantities such as the temperature of actuators or the state the environment around the robot. Consider learning a dynamics model for an aerobatic helicopter. Abbeel et al. (2010) attempted to map only the helicopter's pose and velocity to its acceleration and they found their model to be inaccurate when predicting aggressive maneuvers. They posited that the substantial airflow generated by the helicopter affected the dynamics. Since it is often impossible to directly measure the state of the airflow around a vehicle, identification must be performed in a partially observed setting.

System-identification is a mature field with a rich history (Ljung, 1999; 2010). Various techniques can be classified by whether they apply to linear or non-linear systems, with partially or fully observed states. Additionally, techniques are applied in an online or batch-offline setting. This work presents an approach to offline identification of non-linear and partially observed systems. When a system is fully observed, i.e. its full state is observed but corrupted by noise, a set of techniques called *equation-error methods* are typically employed (Åström & Eykhoff, 1971). In such cases, we can consider observations as independent, and minimize the error between the observed state-derivatives and those predicted by the model given the control input and observed states. In partially observed settings, merely knowing the current input is insufficient to accurately predict the observation. Several black-box approaches exist to predict observations from time-series of inputs. Autoregressive approaches directly map a time-history of past inputs to observations (Billings, 2013). Recurrent neural networks (Bailer-Jones et al., 1998; Zimmermann & Neuneier, 2000) and

subspace-identification methods (Van Overschee & De Moor, 1994) can also be used to learn black-box dynamical systems from this data.

However, in many cases prior knowledge can be used to specify structured, parameterized models of the system (Gupta et al., 2019). Such models can be trained with less data and used with a wider array of control and state-estimation techniques than non-linear black-box models (Gupta et al., 2019; Lutter et al., 2019b;a). Techniques used to identify partially observed structured models are often based on Expectation-Maximization (EM) (Dempster et al., 1977; Schön et al., 2011; Kantas et al., 2015; Ghahramani & Roweis, 1999). An alternating procedure is performed in which a *smoothing* step uses the current system dynamics estimate to infer the distribution over state-trajectories, and is followed by a *learning* step that uses this distribution to update the system dynamics estimate.

In the non-linear or non-Gaussian case, it is typically not possible to analytically characterize the distribution over trajectories, and thus methods based on Sequential Monte-Carlo such as Particle Smoothing (PS) (Schön et al., 2011; Kantas et al., 2015), or Extended Kalman Smoothing (EKS) (Ghahramani & Roweis, 1999) are employed in the E-step. Though considered state-of-the-art for this problem, both methods become intractable in high-dimensional state spaces. PS suffers from the *curse of dimensionality*, requiring an intractably large number of particles if the state space is high-dimensional (Snyder et al., 2008; Kantas et al., 2015), and an M-step that can be quadratic in complexity with respect to the number of particles (Schön et al., 2011). EKS-based methods are fast during the E-step, but the M-step requires approximations to integrate out state uncertainty, such as fitting non-linearities with Radial Basis Function approximators, and scales poorly with the dimension of the state-space (Ghahramani & Roweis, 1999).

In this work, we present a system-identification algorithm that is suited for high-dimensional, non-linear, and partially observed systems. By assuming that the systems are close to deterministic, as is often the case in robotics, we approximate the distribution over unobserved states using only their maximum-likelihood (ML) point-estimate. Our algorithm, called SISL (**S**ystem-identification via **I**terative **S**moothing and **L**earning) performs the following two steps until convergence:

- In the *smoothing* or E-step, we use non-linear programming to tractably find the ML point-estimate of the unobserved states.

- In the *learning* or M-step, we use the estimate of unobserved states to improve the estimate of system parameters.

The idea to use an ML point-estimate in lieu of the distribution over unobserved states in the EM procedure's E-step is not new, and, in general, does not guarantee monotonic convergence to a local optimum (Celeux & Govaert, 1992). However, such an approximation is equivalent to regular EM if the ML point-estimate is the only instance of unobserved variables with non-negligible probability (Celeux & Govaert, 1992; Neal & Hinton, 1998). We apply this idea to the problem of system-identification for nearly deterministic systems, in which ML point-estimates can serve as surrogates for the true distribution over unobserved state-trajectories.

The primary contribution of this work is an algorithm for identifying non-linear, partially observed systems that is able to scale to high-dimensional problems. In Section 2, we specify the assumptions underpinning our algorithm and discuss the computational methodology for using it. In Section 3, we empirically demonstrate that it is able to identify the parameters of a high-dimensional system of coupled Lorenz attractors, a problem that proves intractable for particle-based methods. We also demonstrate our algorithm on the problem of identifying the dynamics of an aerobatic helicopter, and compare against various approaches including the state-of-the-art approach (Punjani & Abbeel, 2015).

## 2 METHODOLOGY

### 2.1 FORMAL PROBLEM STATEMENT

In this work, we assume that we are given a batch of trajectories containing observations $y_{1:T} \in \mathbb{R}^{m \times T}$ of a dynamical system as it evolves over a time horizon $T$, possibly forced by some known input sequence $u_{1:T-1}$. We assume that this dynamical system has a state $x \in \mathbb{R}^n$ that evolves and

generates observations according to the following equations,

$$
\begin{aligned}
x_{t+1} &= f(x_t, u_t, t) + w_t, & w_t &\sim p_w(\cdot) \\
y_t &= g(x_t, u_t, t) + v_t, & v_t &\sim p_v(\cdot)
\end{aligned}
\tag{1}
$$

where $w_t$ is referred to as the *process noise* and $v_t$ as the *observation noise*. $w_t$ and $v_t$ are both assumed to be additive for notational simplicity, but this is not a required assumption. Without loss of generality, we can drop the dependence on $u_t$, absorbing it into the dependence on $t$.

We further assume that we are provided a class of parameterized models $f_\theta(x, t)$ and $g_\theta(x, t)$ for $\theta \in \Theta$ that approximate the dynamical system's evolution and observation processes. The goal of our algorithm is to find the parameters $\theta$ that maximize the likelihood of the observations. That is, we seek to find:

$$
\begin{aligned}
\theta_{\text{ML}} &= \arg\max_\theta p(y_{1:T} \mid \theta) \\
&= \arg\max_\theta \int_{x_{1:T}} p(y_{1:T} \mid x_{1:T}, \theta) p(x_{1:T} \mid \theta) dx_{1:T}
\end{aligned}
\tag{2}
$$

Assuming, the system is Markovian in $x_t$, we can factorize the distributions as:

$$
\begin{aligned}
p(x_{1:T} \mid \theta) &= p(x_1) \prod_{t=1}^{T-1} p_w(x_{t+1} - f_\theta(x_t, t)), \\
p(y_{1:T} \mid x_{1:T}, \theta) &= \prod_{t=1}^{T} p_v(y_t - g_\theta(x_t, t))
\end{aligned}
\tag{3}
$$

In order to tractably solve the maximization problem in Equation (2), particle-based EM techniques typically approximate the integral as an expectation over a particle set (Schön et al., 2011; Kantas et al., 2015). However, since particle-based methods can struggle with high-dimensional spaces (Snyder et al., 2008; Kantas et al., 2015), this work seeks point estimates for the likelihood-maximizing state sequence, which can be found using non-linear programming.

## 2.2 SURROGATE OBJECTIVE FUNCTION

Instead of solving the full maximum likelihood problem shown in Equation (2), we solve a surrogate problem:

$$
\theta_{\text{SISL}} = \arg\max_\theta \left( \max_{x_{1:T}} p(y_{1:T} \mid x_{1:T}, \theta) p(x_{1:T} \mid \theta) \right)
\tag{4}
$$

The maximized surrogate objective is equivalent to the original objective if the distribution over $x_{1:T}$ given $y_{1:T}$ and $\theta$ is well-approximated by its ML point-estimate, i.e. the distribution is a Dirac delta function. We expect this assumption to hold if the trajectories are sufficiently long, the dynamics are close to deterministic, and all relevant dynamic modes are persistently excited. Making these assumptions, we now present an algorithm for solving Equation (4).

## 2.3 SYSTEM-IDENTIFICATION VIA ITERATIVE SMOOTHING AND LEARNING

By taking the logarithm of the likelihood-objective in Equation (4), and using the factorization from Equation (3), we get the SISL objective $J(x_{1:T}, \theta)$ as follows:

$$
J(x_{1:T}, \theta) = \log p(x_1) + \sum_{t=1}^{T} \log p_v(y_t - g_\theta(x_t, t)) + \sum_{t=1}^{T-1} \log p_w(x_{t+1} - f_\theta(x_t, t))
\tag{5}
$$

Jointly maximizing this objective over $x_{1:T}$ and $\theta$ yields $\theta_{\text{SISL}}$.

Though this objective can be optimized as is by a non-linear optimizer, it is not necessarily efficient to do so since $\theta$ and $x_{1:T}$ are highly coupled, leading to inefficient and potentially unstable updates. For this reason, we take an iterative approach akin to EM, which performs block-coordinate ascent on the $J(x_{1:T}, \theta)$.

At iteration $k$, we first perform *smoothing* by holding $\theta$ constant, and finding the ML point-estimate for $x_{1:T}$ as follows:

$$x_{1:T}^{(k)} = \arg\max_{x_{1:T}} J(x_{1:T}, \theta^{(k-1)}) + \rho_x \|x_{1:T} - x_{1:T}^{(k-1)}\|_2^2 \tag{6}$$

Here $\rho_x$ scales a soft trust-region regularizer similar to damping terms found in Levenberg-Marquardt methods (Levenberg, 1944; Marquardt, 1963). The smoothing problem can be solved in $\mathcal{O}(n^2 T)$ by taking advantage of sparsity in the smoothing objective's Hessian matrix.

In the *learning* step, we hold $x_{1:T}$ constant and find:

$$\theta^{(k)} = \arg\max_{\theta} J(x_{1:T}^{(k)}, \theta) + \rho_\theta \|\theta - \theta^{(k-1)}\|_2^2 + \log p(\theta) \tag{7}$$

Again, $\rho_\theta$ scales a soft trust-region regularizer, and specifying $\log p(\theta)$ allows us to regularize $\theta$ toward a prior. The above optimization problem can be solved using any non-linear optimizer, such as a first or second-order gradient descent scheme. The SISL algorithm iterates between the smoothing and learning steps until convergence.

## 3 EXPERIMENTS

The objective of our experiments is to demonstrate that the SISL algorithm is capable of identifying high-dimensional non-linear systems in partially observed settings. We do so in simulation by identifying the parameters of a system of partially observed coupled Lorenz attractors, as well as by identifying the dynamics of a real aerobatic helicopter. In the second experiment, we build on previous analysis of the dataset (Abbeel et al., 2010; Punjani & Abbeel, 2015) by attempting to characterize the interaction of the helicopter with the fluid around it, without having any direct observation of the fluid state.

### 3.1 IDENTIFICATION OF COUPLED LORENZ SYSTEMS

In this didactic experiment, we will show that:

1. SISL learns unbiased parameter estimates of systems that are close to deterministic, and,
2. SISL scales to high-dimensional problems in which particle-smoothing is intractable.

To justify these claims, we use a system that is sufficiently non-linear and partially observable to make particle-based smoothing methods intractable. We choose a system of coupled Lorenz attractors for this purpose, owing to their ability to exhibit chaotic behavior and their use in non-linear atmospheric and fluid flow models (Bergé et al., 1984). Arbitrary increase in state dimensionality can be achieved by coupling multiple individual attractors. The state of a system with $K$ coupled Lorenz attractors is $x \in \mathbb{R}^{3K} = \{\ldots, x_{1,k}, x_{2,k}, x_{3,k}, \ldots\}$. The dynamics of the system are as follows:

$$\begin{aligned}
\dot{\tilde{x}}_{1,k} &= \sigma_k(x_{2,k} - x_{1,k}) \\
\dot{\tilde{x}}_{2,k} &= x_{1,k}(\rho_k - x_{3,k}) - x_{2,k} \\
\dot{\tilde{x}}_{3,k} &= x_{1,k}x_{2,k} - \beta_k x_{3,k} \\
\dot{x} &= \dot{\tilde{x}} + Hx
\end{aligned} \tag{8}$$

where $H$ is an $\mathbb{R}^{3K \times 3K}$ matrix.

We nominally set the parameters $(\sigma_k, \rho_k, \beta_k)$ to the values $(10, 28, 8/3)$, and randomly sample the entries of $H$ from a normal distribution to generate chaotic and coupled behavior between attractors, while avoiding self-coupling. These parameters are estimated during identification. In order to make the system partially observed, the observation $y \in \mathbb{R}^{(3K-2)}$ is found from $x$ as follows:

$$y = Cx + v, \quad v \sim \mathcal{N}(0, \sigma_v^2 \mathbf{I}) \tag{9}$$

where $C \in \mathbb{R}^{(3K-2) \times 3K}$ is a known matrix with full row-rank, and $v$ is the observation noise sampled from a Gaussian with diagonal covariance $\sigma_v^2 \mathbf{I}$. The entries of $C$ are also randomly sampled from a standard normal distribution. In the following experiments, we simulate the system for $T = 128$ timesteps at a sample rate of $\Delta t = 0.04$s, and integrate the system using a $4^{\text{th}}$-order Runge-Kutta method. Initial conditions for each trajectory are sampled such that $x_{1,k} \sim \mathcal{N}(-6, 2.5^2), x_{2,k} \sim \mathcal{N}(-6, 2.5^2), x_{3,k} \sim \mathcal{N}(24, 2.5^2)$.

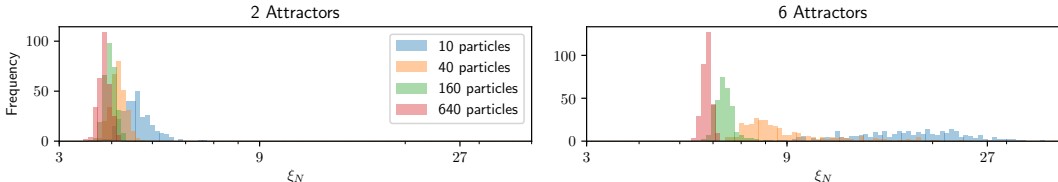

Figure 1: Histograms of weighted RMSE of PS with a varied number of particles.

### 3.1.1 UNBIASED ESTIMATION IN DETERMINISTIC SETTINGS

To test the conditions under which SISL learns unbiased parameter estimates, we simulate a single Lorenz system with $H = 0$, and known $C \in \mathbb{R}^{2 \times 3}$. We introduce and vary the process noise $w \sim \mathcal{N}(0, \sigma_w^2 \mathbf{I})$, and vary the observation noise coefficient $\sigma_v$, and then attempt to estimate the parameters $(\sigma, \rho, \beta)$. Using initial guesses within $10\%$ of the system's true parameter values, we run SISL on a single sampled trajectory. For each choice of $\sigma_w$ and $\sigma_v$, we repeat this process for 10 random seeds.

| $\sigma_w$ | $\sigma_v$ | $\sigma$ | $\rho$ | $\beta$ |
|---|---|---|---|---|
| 0.001 | 0.01 | 10.011(0.012) | 28.000(0.001) | 2.667(0.000) |
| 0.010 | 0.01 | 10.017(0.012) | 28.000(0.001) | 2.668(0.001) |
| 0.100 | 0.01 | 10.064(0.036) | 27.996(0.013) | **2.676 (0.004)** |
| 0.001 | 0.05 | 10.006(0.016) | 27.998(0.002) | 2.666(0.001) |
| 0.001 | 0.10 | 9.998(0.022) | 27.995(0.004) | 2.665(0.001) |

Table 1: Mean parameter estimates and standard errors for a single Lorenz system simulated with various $\sigma_w$ and $\sigma_v$.

Table 1 shows the mean and standard errors of parameter estimates for various $\sigma_w$ and $\sigma_v$. We highlight in red the mean estimates that are not within two standard errors of their true value. We see that $\sigma$ and $\rho$ are estimated without bias for all scenarios. However, the estimate of $\beta$ appears to become biased as the process noise is increased, but not as the observation noise is increased. This supports the assumption that the objective used in SISL is sound when systems evolve close to deterministically, but can be biased if it is not.

### 3.1.2 INTRACTABILITY OF PARTICLE SMOOTHING

State-of-the-art methods for parameter identification of partially observed systems rely on particle-smoothing (PS) to estimate a distribution over states $x_{1:T}$ given a trajectory of observations $y_{1:T}$. We will experimentally demonstrate that the performance of PS does not scale to high-dimensional systems. To do so, we compare the number of particles required for PS to reliably characterize the distribution over $x_{1:T}$ for a system of two and six coupled Lorenz systems. The systems are simulated with observation noise $\sigma_v = 0.01$ but without process noise.

We implement a PS as specified by Schön et al. (2011). Let $\hat{x}_{t,n}, w_{t,n}$ be the estimated system-state and particle weight corresponding to the $n$th particle at the $t$th timestep. In order to test whether PS can reliably characterize the posterior distribution over $x_{1:T}$, we measure the weighted root mean square error (RMSE), $\xi$ as follows:

$$\xi_N = \frac{1}{NT} \sum_{t=1}^{T} \sum_{n=1}^{N} \frac{w_{t,n} \|\hat{x}_{t,n} - x_t\|_2}{\sum_{n=1}^{N} w_{t,n}} \tag{10}$$

If PS is reliable using $N$ particles, we should see that $\xi_N$ is tightly distributed across random seeds.

Figure 1 shows histograms of $\xi_N$ for $N = [10, 40, 160, 640]$. For the two-attractor system, many random seeds appear to have values of $\xi_N$ greater than that of the true posterior state distribution only in the case of $N = 10$. However, in the case of the higher-dimensional six-attractor system, not even 160 particles are sufficient to reliably characterize the true posterior, shown by the fact that the majority of random seeds have large values for $\xi_N$.

These results experimentally demonstrate that, for a sufficiently non-linear and partially observable system, the number of particles required to reliably characterize the posterior distribution over hidden states grows intractably with the dimension of the system. Since particle-based EM methods for system-identification are typically super-linear in complexity with respect to the number of particles (Schön et al., 2011), these methods are ill-suited to high-dimensional problems.

### 3.1.3 CONVERGENCE OF SISL ON HIGH-DIMENSIONAL PROBLEMS

To demonstrate that SISL is capable of identifying high-dimensional systems, we show that we can estimate the dynamics of an $18$ dimensional system of six coupled Lorenz attractors. Moreover, as the number of trajectories provided to SISL increases, it converges to more accurate estimates. To test this claim, we sample $2$, $4$, and $8$ trajectories from a system with parameters $\theta_{\text{true}}$, and $\sigma_v = 0.01$. We randomly initialize each element of the parameters being optimized ($\theta = [\sigma_{1:K}, \rho_{1:K}, \beta_{1:K}, H]$) to within $10\%$ of the their value in $\theta_{\text{true}}$. We then run SISL on each batch, tracking the error in the estimated dynamics as training proceeds. We measure this error, which we call $\epsilon(\theta)$, as follows:

$$\epsilon(\theta) = \mathbf{E}_{x \sim p(x_0)} \left[ \| f_\theta(x) - f_{\theta_{\text{true}}}(x) \|_2 \right] \tag{11}$$

In the learning step, we do not regularize $\theta$ to a prior and set $\rho_\theta = 0$.

In Figure 2, we see the results of this experiment for four random seeds for each batch size. We can see that, as the number of trajectories used in training increases, the error in the estimated dynamics tends toward zero. Furthermore, we see that SISL convergences monotonically to a local optimum in all cases. This experiment supports our claim that SISL is able to identify the parameters of a high-dimensional, non-linear, and partially observable system that is intractable for particle-based methods.

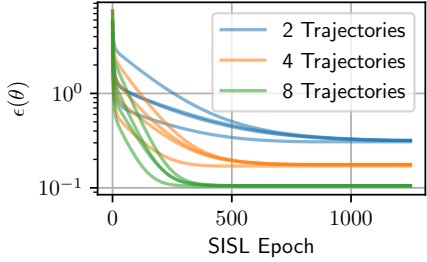

Figure 2: Error in estimated dynamics as SISL trains on a varied number of trajectories from a system of six coupled Lorenz attractors.

The experiments conducted thus far have demonstrated that SISL can learn unbiased parameter estimates of nearly-deterministic systems, and can scale to high-dimensional problems for which particle-based methods are intractable. In the next experiment, we use SISL to characterize the effect of unobserved states on the dynamics of an aerobatic helicopter.

### 3.2 CHARACTERIZING AEROBATIC HELICOPTER DYNAMICS

Characterizing the dynamics of a helicopter undergoing aggressive aerobatic maneuvers is widely considered to be a challenging system-identification problem (Abbeel et al., 2010; Punjani & Abbeel, 2015). The primary challenge is that the forces on the helicopter depend on the induced state of the fluid around it. The state of the fluid cannot be directly observed and its dynamics model is unknown. Merely knowing the state of the helicopter and the control commands at a given time does not contain enough information to accurately predict the forces that act on it.

In order to address this issue, Punjani & Abbeel (2015) use an approach based on Takens theorem, which suggests that a system's state can be reconstructed with a finite number of lagged-observations of it (Takens, 1981). Instead of attempting to estimate the unobserved fluid state, they directly learn a mapping from a $0.5\,\text{s}$ long history of observed state measurements and control commands to the forces acting on the helicopter.

This approach is sensible, and is equivalent to considering the past $0.5\,\text{s}$ of observations as the system's state. However, it can require a very large number of lagged observations to represent complex phenomena. Having such a high dimensional state can make the control design and state-estimation more complicated. To avoid large input dimensions, a trade-off between the duration of the history and sample frequency is necessary. This trade-off will either hurt the resolution of low-frequency content or will alias high-frequencies. We attempt to instead explicitly model the unobserved states affecting the system.

### 3.2.1 OBJECTIVE AND DATASET

The objective of this learning problem is to predict $y_t$, the helicopter's acceleration at time $t$, from an input vector $u_t$ containing the current measured state of the helicopter (its velocity and rotation rates) and the control commands.

We use data collected by the Stanford Autonomous Helicopter Project (Abbeel et al., 2010). Trajectories are split into $10\,\text{s}$ long chunks and then randomly distributed into train, test, and validation sets according to the protocol established by Abbeel et al. (2010); Punjani & Abbeel (2015) and summarized in Appendix A.1. The train, test and validation sets respectively contain 466, 100 and 101 trajectories of 500 time-steps each.

A simple success metric on a given trajectory is the root mean squared prediction error, RMSE = $\sqrt{\frac{1}{T}\sum_{t=1}^{T}\left\|y_t^{(measured)}-y_t^{(pred)}\right\|_2^2}$, where $y_t^{(measured)}$ is the measured force from the dataset, $y_t^{(pred)}$ is the force predicted by the model, and $T$ is the number of time-steps in each trajectory.

### 3.2.2 PREVIOUS WORK AND BASELINES

We first consider a naive baseline that does not attempt to account for the time-varying nature of the fluid-state. We train a neural-network to map only the current helicopter state and control commands to the accelerations: $y_t = \text{NN}_{\theta_n}(u_t)$, where $\text{NN}_{\theta_n}$ is a neural-network with parameters $\theta_n$. We refer to this model as the *naive model*.

We also compare to the work of Punjani & Abbeel (2015). They predict $y_t$ using a time-history $u_{t-H:t}$ of $H$ lagged observations of the helicopter's measured state and control commands. This input is passed through a ReLU-activated neural network with a single hidden-layer combined with what they call a Quadratic Lag Model. As a baseline, we reproduce their performance with a single deep neural network $y_t = \text{NN}_{\theta_h}(u_{t-H:t})$ with parameters $\theta_h$. We call this neural network model the *H25 model*. Both of these models can be trained via stochastic gradient descent to minimize the Mean-Squared-Error (MSE) of their predictions for $y$. The optimization methodology for these models is described in Appendix A.2.

As a third baseline, we compare with subspace-identification methods (Van Overschee & De Moor, 1994). We let $\tilde{y}_t = y_t - \text{NN}_{\theta_n}(u_t)$ be the prediction errors of the trained naive model. We use the MATLAB command `n4sid` to fit a linear dynamical system of the following form:

$$x_{t+1} = A_s x_t + B_s u_t; \qquad \tilde{y}_t = C_s x_t + D_s u_t \tag{12}$$

Here, $x \in R^d$ is the unobserved state with arbitrary dimension $d$. The learned parameters are $\theta_s = [A_s, B_s, C_s, D_s]$. We use a state dimension of 10 and call this model the *SID model*. The `n4sid` algorithm scales super-linearly with the amount of data supplied, and thus we train on 10 randomly sampled subsets of 100 trajectories each, and report the distribution in prediction performance.

Particle-based EM methods are not presented as baselines because they are intractable on problems with large state-spaces, as shown in Section 3.1.2.

### 3.2.3 NON-LINEAR UNOBSERVED STATE MODEL

Similar to the parameterization used for subpace-identification, we fit the prediction errors of the naive model using the following dynamical system:

$$x_{t+1} = A_{\text{NL}} x_t + B_{\text{NL}} u_t; \qquad \tilde{y}_t = C_{\text{NL}} x_t + D_{\text{NL}} u_t + \text{NN}_{\bar{\theta}_{\text{NL}}}(x_t, u_t) \tag{13}$$

where $\text{NN}_{\bar{\theta}_{\text{NL}}}$ is a neural network, and $\theta_{\text{NL}} = [A_{\text{NL}}, B_{\text{NL}}, C_{\text{NL}}, D_{\text{NL}}, \bar{\theta}_{\text{NL}}]$ are the learned parameters. While learning, we assume that both process and observation noise are distributed with diagonal Gaussian covariance matrices $\sigma_w \mathbf{I}$ and $\sigma_v \mathbf{I}$ respectively. The values of $\sigma_w$ and $\sigma_v$ are treated as hyperparmeters of SISL. Here as well, we use a state dimension of 10 and call this model the *NL model*. The optimization methodology for this model is described in Appendix A.2.

It should be noted that the system we learn need not actually correspond to an interpretable model of the fluid-state, but only of time-varying hidden-states that are useful for predicting the accelerations of the helicopter. Expert knowledge of helicopter aerodynamics could be used to further inform a gray-box model trained with SISL.

### 3.2.4 EVALUATION METHODOLOGY

The test RMSE of the *naive* and *H25* models can be evaluated directly on the test trajectories using next-step prediction. However, the *SID* and *NL* models require an estimate of the unobserved state

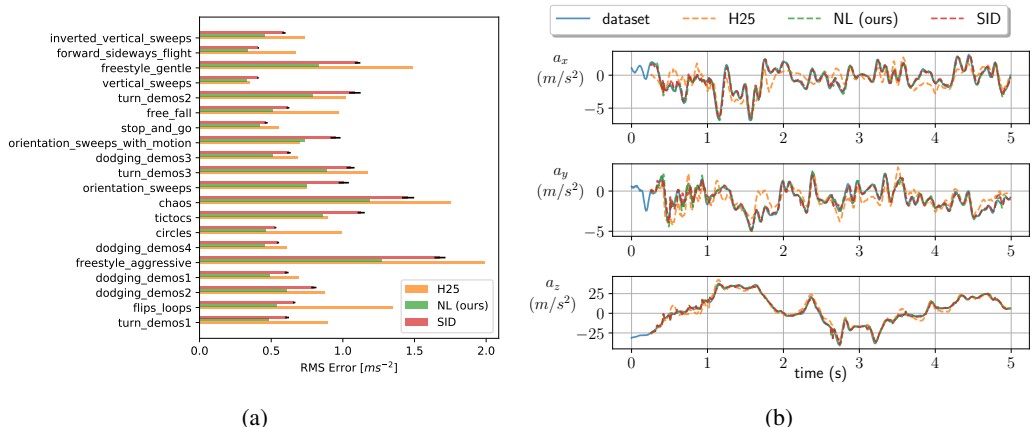

(a)                                                    (b)

Figure 3: (a): Test performance of optimized models on various trajectories. Error bars on *SID* represent the standard deviation of performance of the 10 trained models. (b): Predicted acceleration along axis $x$, $y$ and $z$ in the body frame for one of the harder test set trajectories. Larger versions of these plots, including rotational accelerations, can be found in Appendix A.3.

before making a prediction. The natural analog of next-step prediction is extended Kalman filtering (EKF), during which states are recursively predicted and corrected given observations. At a given time-step, a prediction of $\tilde{y}_t$ is made using the current estimate of $x_t$, and is used in the computation of RMSE. The state-estimate is then corrected with the measured $\tilde{y}_t$.

### 3.2.5 RESULTS

Figure 3a shows the RMSE of the compared models on trajectories in the test-set. We see that the *NL* model is able to consistently predict the accelerations on the helicopter with better accuracy than any of the other models. The naive model performs on average 2.9 times worse than the *H25* model, and its results can be found in Appendix A.3. The *SID* model notably outperforms the state-of-the-art *H25* model, suggesting that a large linear dynamical system can be used to approximate a non-linear and partially observable system (Korda & Mezić, 2018). However, introducing non-linearity as in the *NL* model noticeably improves performance.

Figure 3b depicts the errors in prediction over a sample trajectory in the test-set. Here, we also see that the *NL* model is able to attenuate the time-varying error present in predictions made by the *H25*, suggesting that it has accurately characterized the dynamics of unobserved, time-varying states.

This experiment validates the effectiveness of SISL to identify a non-linear dynamical model of unobserved states that affect the forces acting an aerobatic helicopter.

## 4 CONCLUSIONS

This paper presented an algorithm for system identification of non-linear systems given partial state observations. The algorithm optimizes system parameters given a time history of observations by iteratively finding the most likely state-history, and then using it to optimize the system parameters. The approach is particularly well suited for high-dimensional and nearly deterministic problems.

In simulated experiments on a partially observed system of coupled Lorenz attractors, we showed that our algorithm can perform identification on a problem that particle-based EM methods are fundamentally ill-suited for. We also validated that our algorithm is an effective replacement for identification methods based on EM if the system is close to deterministic, but can yield biased parameter estimates if it is not. We then used our algorithm to model the time-varying hidden-states that affect the dynamics of an aerobatic helicopter. Our approach outperforms state-of-the-art methods because it is able to fit large non-linear models to unobserved states.

We aim to apply our algorithm to system identification problems in a number of domains. There has recently been interest in characterizing the dynamics of aircraft with high aspect ratios, for which the difficult-to-observe bending modes substantially impact dynamics. Additionally, the inability to measure friction forces in dynamic interactions involving contact typically stands in the way of system identification, and thus requires algorithms that are capable of identification under partial observation.

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

# A APPENDIX

## A.1 DATASET COLLECTION AND PREPROCESSING

In this work we use the dataset gathered by Abbeel et al. (2010) and available at `http://heli.stanford.edu/`. A gas-powered helicopter was flown by a professional pilot to collect a large dataset of 6290s of flight. There are four controls: the longitudinal and lateral cyclic pitch, the tail rotor pitch and the collective pitch. The state is measured thanks to an accelerometer, a gyroscope, a magnetometer and vision cameras. Abbeel et al. (2010) provide the raw data, as well as states estimates in the Earth reference frame obtained with extended Kalman smoothing. Following Punjani & Abbeel (2015)'s protocol, we use the fused sensor data and downsample it from 100Hz to 50Hz.

From the Earth frame accelerations provided in the dataset, we compute body frame accelerations (minus gyroscopic terms) which are the prediction targets for our training. Using the notations from Punjani & Abbeel (2015), we can write the helicopter dynamics in the following form:

$$\dot{s} = F(s,\delta) = \begin{bmatrix} C_{12}v \\ \frac{1}{2}\hat{\omega}q \\ C_{12}^\top g - \omega \times v + f_v(s,\delta) \\ f_\omega(s,\delta) \end{bmatrix} \tag{14}$$

where $s \in \mathbb{R}^{13}$ is the helicopter state consisting of its position $r$, quaternion-attitude $q$, linear velocity $v$, angular velocity $\omega$, and $\delta \in \mathbb{R}^4$ to be the control command. $C_{12}$ is the rotation-matrix from the body to Earth reference frame, and $f_v$ and $f_\omega$ are the linear and angular accelerations caused by aerodynamic forces, and are what we aim to predict.

This notation connects with the one used in our paper in the following way:

- We define $u$ as the concatenation of all inputs to the model, including the relevant state variables $v$ and $\omega$ and control commands $\delta$.
- We define $y$ as the output predicted, which would correspond to a concatenation of $f_v$ and $f_\omega$.
- We define $x$ as the vector of unobserved flow states to be estimated and is not present in their model.

## A.2 TRAINING HELICOPTER MODELS

Neural-networks in the *naive* and *H25* models have eight hidden layers of size 32 each, and tanh non-linearities. We optimize these models using an Adam optimizer (Kingma & Ba, 2015) with a harmonic learning rate decay, and mini-batch size of $512$.

The neural network in the *NL* model has two hidden layers of size 32 each, and tanh non-linearity. We train the *NL* model with SISL, using $\rho_x = \rho_\theta = 0.5$, $\sigma_w = \sigma_v = 1.0$, and use an Adam optimizer to optimize Equation (7) in the learning step. The learning rate for dynamics parameters in $\theta_{\text{NL}}$ is $5.0 \times 10^{-4}$ and observation parameters in $\theta_{\text{NL}}$ is $1.0 \times 10^{-3}$. For its relative robustness, we optimize Equation (6) using a non-linear least squares optimizer with a Trust-Region Reflective algorithm (Jones et al., 2001–) in the smoothing step. This step can be solved very efficiently by providing the solver with the block diagonal sparsity pattern of the Jacobian matrix.

To evaluate the test metric, running an EKF is required. The output of an EKF depends on several user-provided parameters:

- $x_0$: value of the initial state
- $\Sigma_0$: covariance of error on initial state
- $Q$: covariance of process noise
- $R$: covariance of observation noise

In this work, we assume that $Q$, $R$ and $\Sigma_0$ are all set to the identity matrix. $x_0$ is assumed to be 0 on all dimensions.

A well-tuned EKF with an inaccurate initial state value converges to accurate estimations in only a few time steps of transient behavior. Since the *H25* model needs 25 past inputs to predict its first output prediction, we drop the first 25 predictions from the EKF when computing RMSE, thereby omitting some of the transient regime.

## A.3   FIGURES

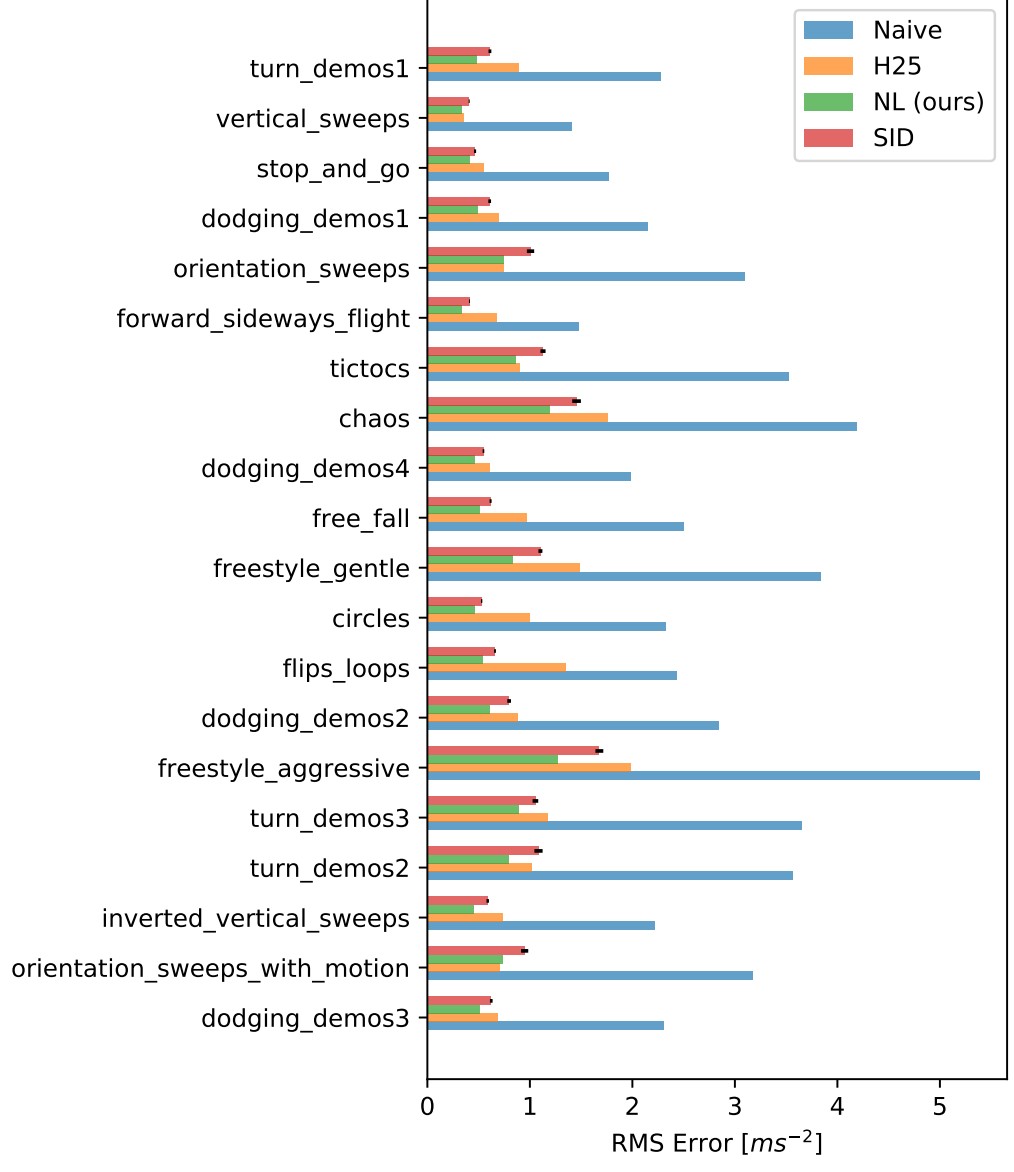

Figure 4: Test performance of optimized models on various trajectories

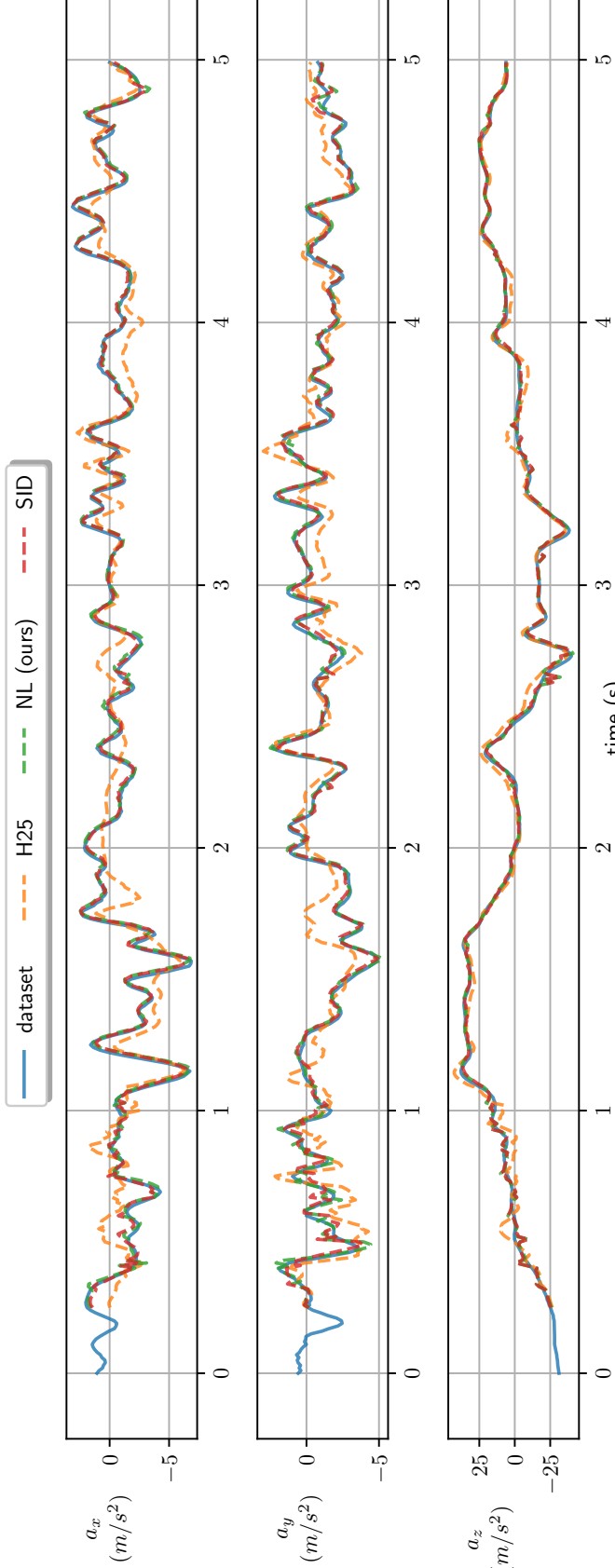

Figure 5: Predicted acceleration along axis $x$, $y$ and $z$ in the body frame. For subspace identification and models trained by SISL, this plot requires running an extended Kalman filter. These figures can be reproduced for any other trajectory with the included code.

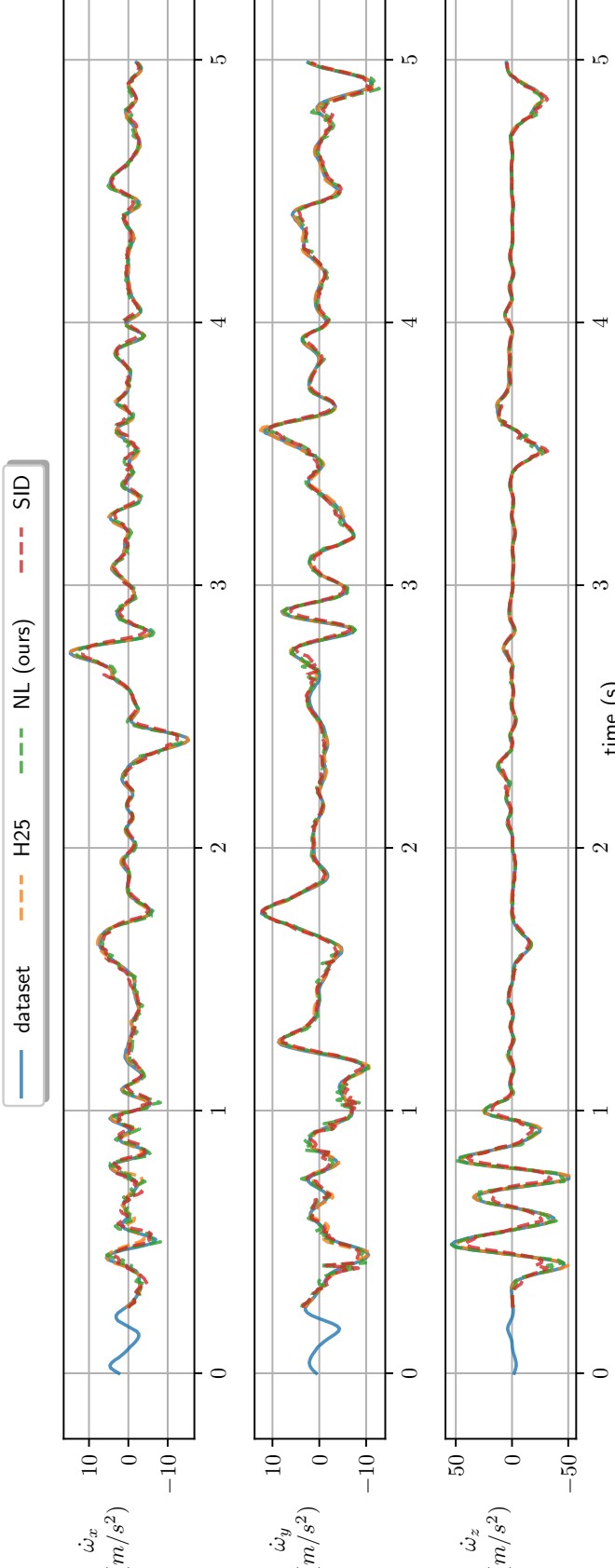

Figure 6: Predicted circular accelerations around axis $x$, $y$ and $z$ in the body frame. For subspace identification and models trained by SISL, this plot requires running an extended Kalman filter. These figures can be reproduced for any other trajectory with the included code.

