# OpenReview forum: "Non-linear System Identification from Partial Observations via Iterative Smoothing and Learning"
_ICLR.cc/2020/Conference — Reject_

### Official Review · AnonReviewer2 · 2019-10-19
**Official Blind Review #2**

**Rating:** 6

**Review:**



#######################
Rebuttal Response:
Thanks for these clarifications and updating the paper. I adapted my score to weak accept. To increase my score to accept, I would like to see evaluations of control experiments rather than just comparing the MSE.

#######################
Review:

Summary:
The paper introduces an iterative learning scheme to perform system identification on partially observable systems. The proposed approach iteratively optimizes the unobserved state and the model parameters. The proposed approach is demonstrated on a 'high'-dimensional Lorenz oscillator and existing quadcopter data. Furthermore, the authors show that existing approaches don't scale to 'high' dimensions.

Conclusion;
The paper introduces a simple iterative scheme to learn a model with a hidden state. The approach is sound and reasonable and the paper is well written but I am uncertain whether this paper fits the scope of ICLR. Maybe R:SS would be a better fit? Furthermore, the experiments can be improved, which is especially important as this paper does not propose a fundamental new approach. It would be really interesting to see this model controlling an actual quadcopter. My current vote is borderline reject but can be improved by further highlighting why quadcopter dynamics are only partially obsevable and further experiments.

Experiments:
My main concern with the paper is the experiments of more complex systems. The initial experiments demonstrating that the approach works and previous methods don't scale to high dimensions are very good and highlight important points. However, the more complex experiments might be not indicative of the performance. Especially for the quadcopter experiments, the proposed method only learns a non-linear noise model while the state transition is still linear and compares it against the purely linear case. Both methods seem qualitatively comparable and the quantitative improvement is only marginal. Therefore, I am doubtful whether the quadcopter data really requires to incorporate the partially observable model and I would prefer to see an evaluation of the proposed iterative scheme using a linear model. Within your evaluation you should compare the different optimization approaches (i.e., naive vs iterative) using the same model representation as your proposed algorithm only proposes a new optimization scheme but is not fixed to a representation. Furthermore, it would be interesting to see a non-linear state model and a non-linear observation model. Does the iterative approach also learn this model? Learning the robot dynamics model with contacts would be the best experiment to validate your approach, as this is only partially observable using the joint measurements and is non-linear. Could you setup an experiment with a simulated two-link robot, generate data containing samples with & without contact and learn the model just using q, qd, qdd, and tau. This experiment would clearly highlight that your iterative scheme can identify a partially observable model.

Minor Comments:
"We let ̃yt=yt−NNθn(ut)be the prediction errors of the trained naive model." -> it is a bit unclear whether the linear model approximates just the noise or the complete model? From the following text I assume that the complete model is modelled as linear model. Then the introduction of the naive model seems unnecessary as the naive model is not used. Could you please clarify this point?

**Experience Assessment:**

I have published one or two papers in this area.

**Review Assessment: Checking Correctness Of Derivations And Theory:**

I carefully checked the derivations and theory.

**Review Assessment: Checking Correctness Of Experiments:**

I carefully checked the experiments.

**Review Assessment: Thoroughness In Paper Reading:**

I read the paper thoroughly.

---

> ### Author Response · Authors · 2019-11-10
> **Response and clarifications to Reviewer #2 - Part 1**
>
> We thank the reviewer for their careful reading and thoughtful suggestions. We address each point individually:
>
> - “whether this paper fits the scope of ICLR”
> We wish to point out two things that our algorithm allows for, when compared to previous approaches:
>
>     + **Learning an augmented state _representation_ that evolves according to some dynamics (linear or non-linear)**: Such representation is very useful from a control and state-estimation perspective as we mention in the Introduction section. Autoregressive black-box models, even when they have better prediction accuracy, can be much harder to use in these applications.
>
>     + **Scaling to large function approximators like neural networks**: Neural networks with large number of parameters have become an essential element in a practitioner’s toolbox to model complex systems from data. However, as we show in this paper, older techniques can fail to scale to the paradigm of large datasets and large number of parameters. For example, SID requires an $O(n^3)$ inversion of matrix in the dataset size $n$ [3].
>  While SISL with its separation of smoothing objective (that can be easily parallelized) and parameter learning approach (that can work with minibatch and first-order gradients) easily scales to this paradigm allowing for improved performance as well as applicability to larger real-world domains.
>
> Given the ICLR Call for Papers includes “learning representations of outputs or states” as subject area [1], and in the past has accepted papers like [2], we believe our work fits within the scope of ICLR and would be of special interest for the robotics and physics modeling community within ICLR.
>
> [1] https://iclr.cc/Conferences/2020/CallForPapers
>
> [2] Lutter, M., Ritter, C., & Peters, J. (2019). Deep lagrangian networks: Using physics as model prior for deep learning. ICLR. URL: https://openreview.net/forum?id=BklHpjCqKm
>
> [3] Qin, S. Joe. "An overview of subspace identification." Computers & chemical engineering 30.10-12 (2006): 1502-1513.

---

> ### Author Response · Authors · 2019-11-10
> **Response and clarifications to Reviewer #2 - Part 2**
>
> - “highlighting why quadcopter dynamics are only partially observable” and the general complexity of the experiments
> It seems the reviewer has a slight misunderstanding about the class of air-vehicles we are dealing with. We wish to clarify that our experiment is performed on data gathered from an _aerobatic helicopter_ and **not** a quadcopter. Unlike a quadcopter flying in nominal conditions, an aerobatic helicopter imparts substantial momentum in the fluid around it, which in turn has a dramatic effect on its dynamics. We point the reviewers to this YouTube video which shows the conditions under which data was collected (https://www.youtube.com/watch?v=VCdxqn0fcnE). The original paper mentions the difficulty created by unobserved fluid states [4], and we baseline against the most recent attempt at tackling it [5]. Furthermore, it is possible to visually confirm that a system is affected by partially observable states by looking at the residuals of a mapping from observed states to accelerations. If there are relevant partially observable states, then errors should be correlated in time. Indeed, this is precisely what we can see in Figure 3b (enlarged in Figure 5), where even using 25 lagged states (H25), predictions are correlated in time, confirming the existence of partially observable states. In the same figure, we see that when using SISL and SID, *with augmented hidden states*, much of the autocorrelation in residuals is accounted for.
>
> - “Learning the robot dynamics model with contacts would be the best experiment to validate your approach”
> The reviewer makes an excellent suggestion to demonstrate our method on a domain involving contact as we mention as our future work. We point out that separation of the smoothing and learning steps makes our method easily applicable to domains with contact. A strength of our approach in this regard is that we can use discrete-mechanics formulations of the smoothing step, thereby explicitly reasoning about normal forces required to support contact constraints. These normal forces can be straightforwardly estimated as partially observed states. However, we believe that the rebuttal window is too short for us to perform the experiment thoroughly and compare against the baseline approaches fairly. However, as we explained in our previous point, the aerobatic-helicopter modeling problem is far from trivial and makes a strong case for the utility of our approach on its own.
>
>
> [4] Abbeel, P., Coates, A., and Ng, A. Y. “Autonomous Helicopter Aerobatics through Apprenticeship Learning.” The International Journal of Robotics Research, Vol. 29, No. 13, 2010, pp. 1608–1639.
> [5] Deep Learning Helicopter Dynamics Models, Ali Punjani, Pieter Abbeel.
> In the proceedings of the IEEE International Conference on Robotics and Automation (ICRA), 2015

---

> ### Author Response · Authors · 2019-11-10
> **Response and clarifications to Reviewer #2 - Part 3**
>
> - The reviewer requests clarity on why we choose a model with linear dynamics and non-linear observations as opposed to one that is fully non-linear. The reason for this is two-fold. Firstly, recent developments in Koopman theory have shown that high-dimensional linear dynamical systems can exactly represent the dynamics of observations that evolve non-linearly with time, provided we use an appropriate non-linear mapping between the high-dimensional linear state and the observation [6]. Given that a state-space model with linear dynamics and non-linear observations is therefore as expressive as one that is fully non-linear, we opt for the former because of the considerable benefits for doing so. Firstly, linear dynamics models since they have fewer local optima, and make the learning problem convex are thus faster to train. Secondly, performing state-estimation and optimal control using such a model are also significantly easier [7].
>
> - This being said, the reviewer notes that the performance gained by using SISL to train a non-linear model with linear dynamics seems marginal compared to simply using SID with a fully linear model. Though the differences do not appear to be huge, they are statistically significant. Furthermore, this finding is to be expected. Non-linear mappings from the hidden-states to the observation are only required to be able represent non-linear dynamical systems with continuous spectrum (i.e. systems without a finite, discrete set of natural frequencies) [6], while finite linear systems always have finite spectra. Large fully linear systems can sometimes do well at representing the main features of a nonlinear system, but need to be excessively large to capture the details of it.
>
> - Clarifications whether the compared approaches “approximates just the noise or the complete model”
> The reviewer has asked for clarification on why we fit the residual errors of the naive model using the NL and SID models, as opposed to directly fitting the target accelerations. The reason for doing so is that the naive model is unable to capture the effects of hidden fluid states, which is precisely what we hope to capture in the NL and SID models.
> We first train a _naive_ model represented by the $\text{NN}_{\theta_n}$ to make acceleration predictions from just the current state. This inevitably leads to temporally-correlated errors in its predictions because it ignores any hidden states in the system. We therefore train another set of linear dynamics and (linear/non-linear) observation models with SID and NL respectively on the errors made by the _naive_ model. This means that the results in Figure 3 refer to predictions from an ensemble of naive + SID and naive + NL respectively. Note that by taking this approach, none of the models are fully linear, and largely explains why we are able to get such good performance from SID. Using tailored models to fit the residuals of coarse models is fairly common in system identification [8].
>
> [6]  Lusch, B., Kutz, J. N., and Brunton, S. L. “Deep Learning for Universal Linear Embeddings of Nonlinear Dynamics.” Nature Communications, Vol. 9, No. 1, 2018, p. 4950. doi:10.1038/s41467-018-07210-0.
>
> [7] Kaiser, Eurika, J. Nathan Kutz, and Steven L. Brunton. "Data-driven discovery of Koopman eigenfunctions for control." arXiv preprint arXiv:1707.01146 (2017).
>
> [8] Ratliff, Nathan, et al. "Doomed: Direct online optimization of modeling errors in dynamics." Big data 4.4 (2016): 253-268.

---

### Official Review · AnonReviewer3 · 2019-10-24
**Official Blind Review #3**

**Rating:** 6

**Review:**

This paper proposes a novel way to address the well studied system identification problem. More specifically, this paper focuses on non linear systems with partial observations. To this aim, the authors propose the "System-indetification via Iterative Smoothing and Learning" (SISL) algorithm. The gist of the approach is to formulate an ML problem to estimate the system model parameters from a parametric family of models to estimate both the dynamical system evolution (state updates) and the observation process. The exact optimization for this ML problem is not tractable. This work proposes to solve a surrogate point-estimate  ML problem. This optimization is solve using an alternate approach (between smoothing and learning).

The paper also presents experimental results both on synthetic  and real data aiming to demonstrate that SISL provides unbiased estimates in deterministic settings. The synthetic setting is also used to motivate the need for methods able to handle large system dimensionality for non-linear and partially observable systems.

The real data setting compares favorably the SISL approach with well established baselines.

I believe that the paper can be improved in the following ways:
1) There are intermediate steps in the derivation of the maths that could be better explained. For example, in equation 2, the “Markovian” aspect of the system is key to derive the formula (x_{t+1} only depends on x_{t}). Equation 4 is not really derived from Equation 3. The factorization shown in Equation 2 is relevant for Equation 4 although the 4) is for the surrogate objective function.

Also, I did not understand why the naive approach is not plotted on Figure 3 (a and b) (but it comes back in the appendix).  Is it just an oversight?

**Experience Assessment:**

I have read many papers in this area.

**Review Assessment: Checking Correctness Of Derivations And Theory:**

I assessed the sensibility of the derivations and theory.

**Review Assessment: Checking Correctness Of Experiments:**

I assessed the sensibility of the experiments.

**Review Assessment: Thoroughness In Paper Reading:**

I read the paper at least twice and used my best judgement in assessing the paper.

---

> ### Author Response · Authors · 2019-11-10
> **Response and clarifications to Reviewer #3**
>
> We thank the reviewer for their helpful and constructive comments. As is correctly pointed out, the system is assumed Markovian in $x_t$ in the derivations of these equations and this factorization allows us to derive Equation 4. We have incorporated these clarifications in our revised manuscript.
>
> Moreover, we wish to clarify that it’s not an oversight that the naive approach is not shown in Fig 3. This was done to ensure the visual clarity of the plots with limited space. As we mention in Section 3.2.5, “naive model performs on average 2.9 times worse than the H25 model” overpowering the distinctions between other approaches when displayed on a small plot.

---

### Official Review · AnonReviewer1 · 2019-10-27
**Official Blind Review #1**

**Rating:** 6

**Review:**

This is a strait forward paper that aims to address the lack of scalability of prior methods on system  identification.  The idea is simple yet the assumptions required are significant and restrict the applicability of the proposed algorithm to systems that are closed to deterministic.  Below are some pros and cons for this paper:

Cons:
(+) The proposed method is simple and it could be  quickly adopted by robotics researchers and practitioners
(+) The method seems to outperform previous algorithms

Pros:
(-) The idea of working with deterministic systems is restrictive.  While the process noise may appear in some robotics applications to be small  I see no way of how one could reduce the effect of the observation noise.    Also in the context of systems that navigate in terrestrial environments  the process noise is definitely more present that in the case of flying vehicles.

(-) It would definitely strengthen the paper if the authors could elaborate more on the concepts of controllability, observability and how these concepts relate to their method etc. How does the existing methods handles these properties? How is the performance of the method affected  when the true system is not controllable but it is stabilizable?


(-) There seems to be no assumption on the shape of the trajectories. What kind of trajectories you need to perform system identification? Can you mathematically represent these assumptions? In the adaptive control literature there exists conditions related to "sufficient rich signals" see the book by Ioannou on Robust Adaptive Control. These are signals that can excite dynamics so that to ensure proper identification.





**Experience Assessment:**

I have published one or two papers in this area.

**Review Assessment: Checking Correctness Of Derivations And Theory:**

I assessed the sensibility of the derivations and theory.

**Review Assessment: Checking Correctness Of Experiments:**

I assessed the sensibility of the experiments.

**Review Assessment: Thoroughness In Paper Reading:**

I read the paper at least twice and used my best judgement in assessing the paper.

---

> ### Author Response · Authors · 2019-11-10
> **Response and clarifications to Reviewer #1**
>
> We thank the reviewer for their constructive feedback.
>
> The reviewer correctly points out that many real world systems do evolve with non-negligible process noise. Still, we believe that there is a large class of important physical and robotic domains such as aerial vehicles where this assumption of low process noise is valid. Moreover, we make no assumptions regarding the observation noise and wish to emphasize that our results from Section 3.1.1 and Table 1 show that observation noise does not introduce bias in the identified parameters.
>
> The reviewer requests some clarity on the connection to controllability and observability. Since our algorithm is applicable to an unrestricted class of non-linear models, it is difficult to make general claims about these properties. We expect that our method can only converge to observable systems as solutions since it explicitly performs state-estimation during identification. However, we do not expect our method to always yield controllable or stablizable systems. Constraints could be added during the learning step, such as those proposed by Singh, et al. [1] which limit the class of models to only represent stabilizable systems.
>
> It is absolutely correct that learning a dynamical system requires the assumption that the dataset is informative enough. In linear system identification and linear adaptive control, the problem can be posed formally by reasoning about the frequency content of the input: all frequencies should have been sufficiently excited in a recent enough history. For nonlinear systems, this is an active area of research [2,3] and there is no general rule without assuming a specific model structure. The helicopter aerobatics dataset was chosen because it is informative enough yet notoriously hard to learn the underlying system, which allowed to evaluate our algorithm’s performance against baselines on a fair basis.
>
> [1] Singh, S., Richards, S. M., Sindhwani, V., Slotine, J. J. E., & Pavone, M. (2019). Learning stabilizable nonlinear dynamics with contraction-based regularization. arXiv preprint arXiv:1907.13122. URL: https://arxiv.org/abs/1907.13122
>
> [2] K. Mahata, J. Schoukens, and A. De Cock. Information matrix and d-optimal design with gaussian inputs for wiener model identification. Automatica, 69:65–77, 2016.
>
> [3] P. E. Valenzuela, C. R. Rojas and H. Hjalmarsson, "Optimal input design for non-linear dynamic systems: A graph theory approach," 52nd IEEE Conference on Decision and Control, Florence, 2013, pp. 5740-5745.

---

### Author Response · Authors · 2019-11-13
**Updated manuscript**

We have uploaded an updated version of the manuscript incorporating the suggestions made by the reviewers. We thank the reviewers again for their constructive feedback and request them to update their evaluation taking into account our responses and revision.

---

### Decision · Program_Chairs · 2019-12-19

**Decision:**

Reject

**Comment:**

The paper is about nonlinear system identification in an EM-style learning framework. The idea is to use nonlinear programming for the E step (finding a MAP estimate) and then refine the model parameters. In flavor, this approach is similar to the work by Roweis and Ghahramani.

However, this paper does not offer any new insights whatsoever and the (very short) methods section arrives at proposing to compute the maximum a posteriori estimate (eq. 5). While the motivation for this given in the paper is a bit hard to understand it is of course a very well-known and useful estimator. Besides the maximum likelihood estimator this is one of the most commonly used point estimators, see any textbook on statistical signal processing. There has been quite a bit of work in the signal processing community over the last 10 years, and a good overview can be found here:
https://web.stanford.edu/~boyd/papers/pdf/rt_cvx_sig_proc.pdf
This should give evidence that this is indeed a standard way of solving the problem and it does work really well. Given that we have so fast and good optimizers these days it is common to solve Kalman filtering/smoothing problems via this optimization problem.
The paper does not contain any analysis at all. The experiments do of course show that the method works (when there is low noise). Again, we know very well that the MAP estimate is a decent estimator for unimodal problems. The MAP estimator can also be made to work well for noisy situations.

As for the comments that the sequential Monte Carlo methods do not work in higher dimensions that is indeed true. However, there are now algorithms that work in much higher dimensions than those considered by the authors of this paper, e.g.
https://ieeexplore.ieee.org/document/8752074
which also contains an up-to-date survey on the topic. Furthermore, when it comes to particle smoothing there are also much more efficient smoothers than 10 years ago. The area of particle smoothing has also evolved rapidly over the past years.

Summary:
The paper makes use of the well-known MAP estimator for learning nonlinear dynamical systems (states and parameters). This is by now a standard technique in signal processing. There are several throw-away comments on SMC that are not valid and that are not grounded in the intense research of that field over the past decade.